# RNADE: The real-valued neural autoregressive density-estimator

**Benigno Uria** and **Iain Murray**
School of Informatics
University of Edinburgh
{b.uria,i.murray}@ed.ac.uk

**Hugo Larochelle**
Département d'informatique
Université de Sherbrooke
hugo.larochelle@usherbrooke.ca

## Abstract

We introduce RNADE, a new model for joint density estimation of real-valued vectors. Our model calculates the density of a datapoint as the product of one-dimensional conditionals modeled using mixture density networks with shared parameters. RNADE learns a distributed representation of the data, while having a tractable expression for the calculation of densities. A tractable likelihood allows direct comparison with other methods and training by standard gradient-based optimizers. We compare the performance of RNADE on several datasets of heterogeneous and perceptual data, finding it outperforms mixture models in all but one case.

## 1 Introduction

Probabilistic approaches to machine learning involve modeling the probability distributions over large collections of variables. The number of parameters required to describe a general discrete distribution grows exponentially in its dimensionality, so some structure or regularity must be imposed, often through graphical models [e.g. 1]. Graphical models are also used to describe probability densities over collections of real-valued variables.

Often parts of a task-specific probabilistic model are hard to specify, and are learned from data using generic models. For example, the natural probabilistic approach to image restoration tasks (such as denoising, deblurring, inpainting) requires a multivariate distribution over uncorrupted patches of pixels. It has long been appreciated that large classes of densities can be estimated consistently by kernel density estimation [2], and a large mixture of Gaussians can closely represent any density. In practice, a parametric mixture of Gaussians seems to fit the distribution over patches of pixels and obtains state-of-the-art restorations [3]. It may not be possible to fit small image patches significantly better, but alternative models could further test this claim. Moreover, competitive alternatives to mixture models might improve performance in other applications that have insufficient training data to fit mixture models well.

Restricted Boltzmann Machines (RBMs), which are undirected graphical models, fit samples of binary vectors from a range of sources better than mixture models [4, 5]. One explanation is that RBMs form a distributed representation: many hidden units are active when explaining an observation, which is a better match to most real data than a single mixture component. Another explanation is that RBMs *are* mixture models, but the number of components is exponential in the number of hidden units. Parameter tying among components allows these more flexible models to generalize better from small numbers of examples. There are two practical difficulties with RBMs: the likelihood of the model must be approximated, and samples can only be drawn from the model approximately by Gibbs sampling. The Neural Autoregressive Distribution Estimator (NADE) overcomes these difficulties [5]. NADE is a directed graphical model, or feed-forward neural network, initially derived as an approximation to an RBM, but then fitted as a model in its own right.

In this work we introduce the Real-valued Autoregressive Density Estimator (RNADE), an extension of NADE. An autoregressive model expresses the density of a vector as an ordered product of one-dimensional distributions, each conditioned on the values of previous dimensions in the (perhaps arbitrary) ordering. We use the parameter sharing previously introduced by NADE, combined with mixture density networks [6], an existing flexible approach to modeling real-valued distributions with neural networks. By construction, the density of a test point under RNADE is cheap to compute, unlike RBM-based models. The neural network structure provides a flexible way to alter the mean and variance of a mixture component depending on context, potentially modeling non-linear or heteroscedastic data with fewer components than unconstrained mixture models.

## 2 Background: Autoregressive models

Both NADE [5] and our RNADE model are based on the chain rule (or product rule), which factorizes any distribution over a vector of variables into a product of terms: $p(\boldsymbol{x}) = \prod_{d=1}^{D} p(x_d \mid \boldsymbol{x}_{<d})$, where $\boldsymbol{x}_{<d}$ denotes all attributes preceding $x_d$ in a fixed arbitrary ordering of the attributes. This factorization corresponds to a Bayesian network where every variable is a parent of all variables after it. As this model assumes no conditional independences, it says nothing about the distribution in itself. However, the (perhaps arbitrary) ordering we choose will matter if the form of the conditionals is constrained. If we assume tractable parametric forms for each of the conditional distributions, then the joint distribution can be computed for any vector, and the parameters of the model can be locally fitted to a penalized maximum likelihood objective using any gradient-based optimizer.

For binary data, each conditional distribution can be modeled with logistic regression, which is called a fully visible sigmoid belief network (FVSBN) [7]. Neural networks can also be used for each binary prediction task [8]. The neural autoregressive distribution estimator (NADE) also uses neural networks for each conditional, but with parameter sharing inspired by a mean-field approximation to Restricted Boltzmann Machines [5]. In detail, each conditional is given by a feed-forward neural network with one hidden layer, $\boldsymbol{h}_d \in \mathbb{R}^H$:

$$p(x_d = 1 \,|\, \boldsymbol{x}_{<d}) = \text{sigm}\left(\boldsymbol{v}_d^\top \boldsymbol{h}_d + b_d\right) \quad \text{where} \quad \boldsymbol{h}_d = \text{sigm}\left(\boldsymbol{W}_{\cdot, <d}\boldsymbol{x}_{<d} + \boldsymbol{c}\right), \quad (1)$$

where $\boldsymbol{v}_d \in \mathbb{R}^H$, $b_d \in \mathbb{R}$, $\boldsymbol{c} \in \mathbb{R}^H$, and $\boldsymbol{W} \in \mathbb{R}^{H \times (D-1)}$ are neural network parameters, and sigm represents the logistic sigmoid function $1/(1 + e^{-x})$.

The weights between the inputs and the hidden units for each neural network are tied: $\boldsymbol{W}_{\cdot, <d}$ is the first $d-1$ columns of a shared weight matrix $\boldsymbol{W}$. This parameter sharing reduces the total number of parameters from quadratic in the number of input dimensions to linear, lessening the need for regularisation. Computing the probability of a datapoint can also be done in time linear in dimensionality, $O(DH)$, by sharing the computation when calculating the hidden activation of each neural network ($\boldsymbol{a}_d = \boldsymbol{W}_{\cdot, <d}\boldsymbol{x}_{<d} + \boldsymbol{c}$):

$$\boldsymbol{a}_1 = \boldsymbol{c}, \qquad \boldsymbol{a}_{d+1} = \boldsymbol{a}_d + x_d \boldsymbol{W}_{\cdot, d}. \quad (2)$$

When approximating Restricted Boltzmann Machines, the output weights $\{\boldsymbol{v}_d\}$ in (1) were originally tied to the input weights $\boldsymbol{W}$. Untying these weights gave better statistical performance on a range of tasks, with negligible extra computational cost [5].

NADE has recently been extended to count data [9]. The possibility of extending generic neural autoregressive models to continuous data has been mentioned [8, 10], but has not been previously explored to our knowledge. An autoregressive mixture of experts with scale mixture model experts has been developed as part of a sophisticated multi-resolution model specifically for natural images [11]. In more general work, Gaussian processes have been used to model the conditional distributions of a fully visible Bayesian network [12]. However, these 'Gaussian process networks' cannot deal with multimodal conditional distributions or with large datasets (currently $\gtrsim 10^4$ points would require further approximation). In the next section we propose a more flexible and scalable approach.

## 3 Real-valued neural autoregressive density estimators

The original derivation of NADE suggests deriving a real-valued version from a mean-field approximation to the conditionals of a Gaussian-RBM. However, we discarded this approach because the

limitations of the Gaussian-RBM are well documented [13, 14]: its isotropic conditional noise model does not give competitive density estimates. Approximating a more capable RBM model, such as the mean-covariance RBM [15] or the spike-and-slab RBM [16], might be a fruitful future direction.

The main characteristic of NADE is the tying of its input-to-hidden weights. The output layer was 'untied' from the approximation to the RBM to give the model greater flexibility. Taking this idea further, we add more parameters to NADE to represent each one-dimensional conditional distribution with a mixture of Gaussians instead of a Bernoulli distribution. That is, the outputs are mixture density networks [6], with a shared hidden layer, using the same parameter tying as NADE.

Thus, our *Real-valued Neural Autoregressive Density-Estimator* or RNADE model represents the probability density of a vector as:

$$p(\boldsymbol{x}) = \prod_{d=1}^{D} p(x_d | \boldsymbol{x}_{<d}) \quad \text{with} \quad p(x_d | \boldsymbol{x}_{<d}) = p_{\mathcal{M}}(x_d | \boldsymbol{\theta}_d), \tag{3}$$

where $p_{\mathcal{M}}$ is a mixture of Gaussians with parameters $\boldsymbol{\theta}_d$. The mixture model parameters are calculated using a neural network with all of the preceding dimensions, $\boldsymbol{x}_{<d}$, as inputs. We now give the details.

RNADE computes the same hidden unit activations, $\boldsymbol{a}_d$, as before using (2). As discussed by Bengio [10], as an RNADE (or a NADE) with sigmoidal units progresses across the input dimensions $d \in \{1 \dots D\}$, its hidden units will tend to become more and more saturated, due to their input being a weighted sum of an increasing number of inputs. Bengio proposed alleviating this effect by rescaling the hidden units' activation by a free factor $\rho_d$ at each step, making the hidden unit values

$$\boldsymbol{h}_d = \text{sigm} \left( \rho_d \boldsymbol{a}_d \right). \tag{4}$$

Learning these extra rescaling parameters worked slightly better, and all of our experiments use them.

Previous work on neural networks with real-valued outputs has found that rectified linear units can work better than sigmoidal non-linearities [17]. The hidden values for rectified linear units are:

$$\boldsymbol{h}_d = \begin{cases} \rho_d \boldsymbol{a}_d & \text{if } \rho_d \boldsymbol{a}_d > 0 \\ 0 & \text{otherwise.} \end{cases} \tag{5}$$

In preliminary experiments we found that these hidden units worked better than sigmoidal units in RNADE, and used them throughout (except for an example result with sigmoidal units in Table 2).

Finally, the mixture of Gaussians parameters for the $d$-th conditional, $\boldsymbol{\theta}_d = \{\boldsymbol{\alpha}_d, \boldsymbol{\mu}_d, \boldsymbol{\sigma}_d\}$, are set by:

$$K \text{ mixing fractions,} \qquad \boldsymbol{\alpha}_d = \text{softmax} \left( \boldsymbol{V}_d^{\alpha\top} \boldsymbol{h}_d + \boldsymbol{b}_d^{\alpha} \right) \tag{6}$$

$$K \text{ component means,} \qquad \boldsymbol{\mu}_d = \boldsymbol{V}_d^{\mu\top} \boldsymbol{h}_d + \boldsymbol{b}_d^{\mu} \tag{7}$$

$$K \text{ component standard deviations,} \qquad \boldsymbol{\sigma}_d = \exp \left( \boldsymbol{V}_d^{\sigma\top} \boldsymbol{h}_d + \boldsymbol{b}_d^{\sigma} \right), \tag{8}$$

where free parameters $\boldsymbol{V}_d^{\alpha}, \boldsymbol{V}_d^{\mu}, \boldsymbol{V}_d^{\sigma}$ are $H \times K$ matrices, and $\boldsymbol{b}_d^{\alpha}, \boldsymbol{b}_d^{\mu}, \boldsymbol{b}_d^{\sigma}$ are vectors of size $K$. The *softmax* [18] ensures the mixing fractions are positive and sum to one, the exponential ensures the standard deviations are positive.

Fitting an RNADE can be done using gradient ascent on the model's likelihood given a training set of examples. We used minibatch stochastic gradient ascent in all our experiments. In those RNADE models with MoG conditionals, we multiplied the gradient of each component mean by its standard deviation (for a Gaussian, Newton's method multiplies the gradient by its variance, but empirically multiplying by the standard deviation worked better). This gradient scaling makes tight components move more slowly than broad ones, a heuristic that we found allows the use of higher learning rates.

**Variants:** Using a mixture of Gaussians to represent the conditional distributions in RNADE is an arbitrary parametric choice. Given several components, the mixture model can represent a rich set of skewed and multimodal distributions with different tail behaviors. However, other choices could be appropriate in particular circumstances. For example, work on natural images often uses scale mixtures, where components share a common mean. Conditional distributions of perceptual data are often assumed to be Laplacian [e.g. 19]. We call our main variant with mixtures of Gaussians RNADE-MoG, but also experiment with mixtures of Laplacian outputs, RNADE-MoL.

Table 1: Average test-set log-likelihood per datapoint for 4 different models on five UCI datasets. Performances not in bold can be shown to be significantly worse than at least one of the results in bold as per a paired $t$-test on the ten mean-likelihoods, with significance level 0.05.

| Dataset | dim | size | Gaussian | MFA | FVBN | RNADE-MoG | RNADE-MoL |
|---|---|---|---|---|---|---|---|
| Red wine | 11 | 1599 | $-13.18$ | $-10.19$ | $-11.03$ | $\mathbf{-9.36}$ | $\mathbf{-9.46}$ |
| White wine | 11 | 4898 | $-13.20$ | $-10.73$ | $-10.52$ | $\mathbf{-10.23}$ | $-10.38$ |
| Parkinsons | 15 | 5875 | $-10.85$ | $-1.99$ | $\mathbf{-0.71}$ | $\mathbf{-0.90}$ | $-2.63$ |
| Ionosphere | 32 | 351 | $-41.24$ | $-17.55$ | $-26.55$ | $\mathbf{-2.50}$ | $\mathbf{-5.87}$ |
| Boston housing | 10 | 506 | $-11.37$ | $-4.54$ | $\mathbf{-3.41}$ | $\mathbf{-0.64}$ | $-4.04$ |

## 4    Experiments

We compared RNADE to mixtures of Gaussians (MoG) and factor analyzers (MFA), which are surprisingly strong baselines in some tasks [20, 21]. Given the known poor performance of discrete mixtures [4, 5], we limited our experiments to modeling continuous attributes. However it would be easy to include both discrete and continuous variables in a NADE-like architecture.

### 4.1    Low-dimensional data

We first considered five UCI datasets [22], previously used to study the performance of other density estimators [23, 20]. These datasets have relatively low dimensionality, with between 10 and 32 attributes, but have hard thresholds and non-linear dependencies that may make it difficult to fit mixtures of Gaussians or factor analyzers.

Following Tang *et al.* [20], we eliminated discrete-valued attributes and an attribute from every pair with a Pearson correlation coefficient greater than 0.98. Each dimension of the data was normalized by subtracting its training subset sample mean and dividing by its standard deviation. All results are reported on the normalized data.

As baselines we fitted full-covariance Gaussians and mixtures of factor analysers. To measure the performance of the different models, we calculated their log-likelihood on held-out test data. Because these datasets are small, we used 10-folds, with 90% of the data for training, and 10% for testing.

We chose the hyperparameter values for each model by doing per-fold cross-validation; using a ninth of the training data as validation data. Once the hyperparameter values had been chosen, we trained each model using all the training data (including the validation data) and measured its performance on the 10% of held-out testing data. In order to avoid overfitting, we stopped the training after reaching a training likelihood higher than the one obtained on the best validation-wise iteration of the corresponding validation run. Early stopping is crucial to avoid overfitting the RNADE models. It also improves the results of the MFAs, but to a lesser degree.

The MFA models were trained using the EM algorithm [24, 25], the number of components and factors were crossvalidated. The number of factors was chosen from even numbers from $2 \ldots D$, where selecting $D$ gives a mixture of Gaussians. The number of components was chosen among all even numbers from $2 \ldots 50$ (crossvalidation always selected fewer than $50$ components).

RNADE-MoG and RNADE-MoL models were fitted using minibatch stochastic gradient descent, using minibatches of size 100, for 500 epochs, each epoch comprising 10 minibatches. For each experiment, the number of hidden units (50), the non-linear activation-function of the hidden units (RLU), and the form of the conditionals were fixed. Three hyperparameters were crossvalidated using grid-search: the number of components on each one-dimensional conditional was chosen from the set $\{2, 5, 10, 20\}$; the weight-decay (used only to regularize the input to hidden weights) from the set $\{2.0, 1.0, 0.1, 0.01, 0.001, 0\}$; and the learning rate from the set $\{0.1, 0.05, 0.025, 0.0125\}$. Learning-rates were decreased linearly to reach 0 after the last epoch.

We also trained fully-visible Bayesian networks (FVBN), an autoregressive model where each one-dimensional conditional is modelled by a separate mixture density network using no parameter tying.

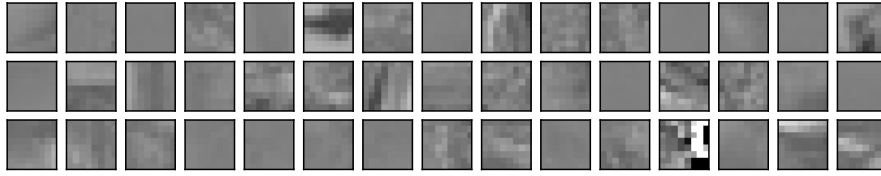

Figure 1: **Top:** 15 8x8 patches from the BSDS test set. **Center:** 15 samples from Zoran and Weiss's MoG model with 200 components. **Bottom:** 15 samples from an RNADE with 512 hidden units and 10 output components per dimension. All data and samples were drawn randomly.

The same cross-validation procedure and hyperparameters as for RNADE training were used. The best validationwise MDN for each one-dimensional conditional was chosen.

The results are shown in Table 1. Autoregressive methods obtained statistical performances superior to mixture models on all datasets. An RNADE with mixture of Gaussian conditionals was among the statistically significant group of best models on all datasets. Unfortunately we could not reproduce the data-folds used by previous work, however, our improvements are larger than those demonstrated by a deep mixture of factor analyzers over standard MFA [20].

## 4.2    Natural image patches

We also measured the ability of RNADE to model small patches of natural images. Following the recent work of Zoran and Weiss [3], we use 8-by-8-pixel patches of monochrome natural images, obtained from the BSDS300 dataset [26] (Figure 1 gives examples).

Pixels in this dataset can take a finite number of brightness values ranging from 0 to 255. Modeling discretized data using a real-valued distribution can lead to arbitrarily high density values, by locating narrow high density spike on each of the possible discrete values. In order to avoid this 'cheating' solution, we added noise uniformly distributed between 0 and 1 to the value of each pixel. We then divided by 256, making each pixel take a value in the range $[0, 1]$.

In previous experiments, Zoran and Weiss [3] subtracted the mean pixel value from each patch, reducing the dimensionality of the data by one: the value of any pixel could be perfectly predicted as minus the sum of all other pixel values. However, the original study still used a mixture of full-covariance 64-dimensional Gaussians. Such a model could obtain arbitrarily high model likelihoods, so unfortunately the likelihoods reported in previous work on this dataset [3, 20] are difficult to interpret. In our preliminary experiment using RNADE, we observed that if we model the 64-dimensional data, the 64th pixel is always predicted by a very thin spike centered at its true value. The ability of RNADE to capture this spurious dependency is reassuring, but we wouldn't want our results to be dominated by it. Recent work by Zoran and Weiss [21], projects the data on the leading 63 eigenvectors of each component, when measuring the model likelihood [27]. For comparison amongst a range of methods, we advocate simply discarding the 64th (bottom-right) pixel.

We trained our model using patches drawn randomly from 180 images in the training subset of BSDS300. A validation dataset containing 1,000 random patches from the remaining 20 images in the training subset were used for early-stopping when training RNADE. We measured the performance of each model by measuring their log-likelihood on one million patches drawn randomly from the test subset, which is composed of 100 images not present in the training subset. Given the larger scale of this dataset, hyperparameters of the RNADE and MoG models were chosen manually using the performance of preliminary runs on the validation data, rather than by an extensive search.

The RNADE model had 512 rectified-linear hidden units and a mixture of 20 one-dimensional Gaussian components per output. Training was done by minibatch gradient descent, with 25 datapoints per minibatch, for a total of 200 epochs, each comprising 1,000 minibatches. The learning-rate was scheduled to start at 0.001 and linearly decreased to reach 0 after the last epoch. Gradient momentum with momentum factor 0.9 was used, but initiated at the beginning of the second epoch. A weight decay rate of 0.001 was applied to the input-to-hidden weight matrix only. Again, we found that multiplying the gradient of the mean output parameters by the standard deviation improves results. RNADE training was early stopped but didn't show signs of overfitting. We produced a further run

Table 2: Average per-example log-likelihood of several mixture of Gaussian and RNADE models, with mixture of Gaussian (MoG) or mixture of Laplace (MoL) conditionals, on 8-by-8 patches of natural images. These results are measured in nats and were calculated using one million patches. Standard errors due to the finite test sample size are lower than $0.1$ in every case. $K$ gives the number of one-dimensional components for each conditional in RNADE, and the number of full-covariance components for MoG.

| Model | Training LogL | Test LogL |
|---|---|---|
| MoG $K = 200$ (Z&W) | 161.9 | **152.8** |
| MoG $K = 100$ | 152.8 | 144.7 |
| MoG $K = 200$ | 159.3 | 150.4 |
| MoG $K = 300$ | 159.3 | 150.4 |
| RNADE-MoG $K = 5$ | 158.0 | 149.1 |
| RNADE-MoG $K = 10$ | 160.0 | 151.0 |
| RNADE-MoG $K = 20$ | 158.6 | 149.7 |
| RNADE-MoL $K = 5$ | 150.2 | 141.5 |
| RNADE-MoL $K = 10$ | 149.7 | 141.1 |
| RNADE-MoL $K = 20$ | 150.1 | 141.5 |
| RNADE-MoG $K = 10$ (sigmoid h. units) | 155.1 | 146.4 |
| RNADE-MoG $K = 10$ (1024 units, 400 epochs) | 161.1 | 152.1 |

with 1024 hidden units for 400 epochs, with still no signs of overfitting; even larger models might perform better.

The MoG model was trained using minibatch EM, for 1,000 iterations. At each iteration 20,000 randomly sampled datapoints were used in an EM update. A step was taken from the previous mixture model towards the parameters resulting from the M-step: $\boldsymbol{\theta_t} = (1 - \eta)\boldsymbol{\theta_{t-1}} + \eta\boldsymbol{\theta_{EM}}$, where the step size ($\eta$) was scheduled to start at 0.1 and linearly decreased to reach 0 after the last update. The training of the MoG was also early-stopped and also showed no signs of overfitting.

The results are shown in Table 2. We compare RNADE with a mixtures of Gaussians model trained on 63 pixels, and with a MoG trained by Zoran and Weiss (downloaded from Daniel Zoran's website) from which we removed the 64th row and column of each covariance matrix. The best RNADE test log-likelihood is, on average, 0.7 nats per patch lower than Zoran and Weiss's MoG, which had a different training procedure than our mixture of Gaussians.

Figure 1 shows a few examples from the test set, and samples from the MoG and RNADE models. Some of the samples from RNADE are unnaturally noisy, with pixel values outside the legal range (see fourth sample from the right in Figure 1). If we constrain the pixels values to a unit range, by rejection sampling or otherwise, these artifacts go away. Limiting the output range of the model would also improve test likelihood scores slightly, but not by much: log-likelihood does not strongly penalize models for putting a small fraction of probability mass on 'junk' images.

All of the results in this section were obtained by fitting the pixels in a raster-scan order. Perhaps surprisingly, but consistent with previous results on NADE [5] and by Frey [28], randomizing the order of the pixels made little difference to these results. The difference in performance was comparable to the differences between multiple runs with the same pixel ordering.

### 4.3 Speech acoustics

We also measured the ability of RNADE to model small patches of speech spectrograms, extracted from the TIMIT dataset [29]. The patches contained 11 frames of 20 filter-banks plus energy; totaling 231 dimensions per datapoint. These filter-bank encoding is common in speech-recognition, and better for visualization than the more frequently used MFCC features. A good generative model of speech could be used, for example, in denoising, or speech detection tasks.

We fitted the models using the standard TIMIT training subset, and compared RNADE with a MoG by measuring their log-likelihood on the complete TIMIT core-test dataset.

Table 3: Log-likelihood of several MoG and RNADE models on the core-test set of TIMIT measured in nats. Standard errors due to the finite test sample size are lower than 0.3 nats in every case. RNADE obtained a higher (better) log-likelihood.

| Model | Training LogL | Test LogL |
|---|---|---|
| MoG $N = 50$ | 111.6 | 110.4 |
| MoG $N = 100$ | 113.4 | 112.0 |
| MoG $N = 200$ | 113.9 | 112.5 |
| MoG $N = 300$ | 114.1 | 112.5 |
| RNADE-MoG $K = 10$ | 125.9 | 123.9 |
| RNADE-MoG $K = 20$ | 126.7 | **124.5** |
| RNADE-MoL $K = 10$ | 120.3 | 118.0 |
| RNADE-MoL $K = 20$ | 122.2 | 119.8 |

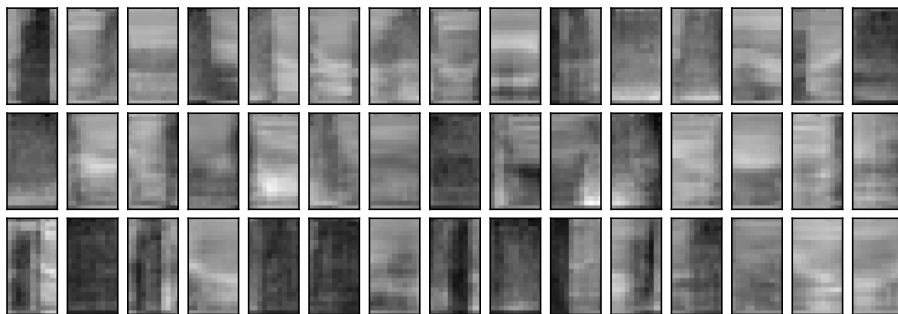

Figure 2: **Top:** 15 datapoints from the TIMIT core-test set. **Center:** 15 samples from a MoG model with 200 components. **Bottom:** 15 samples from an RNADE with 1024 hidden units and output components per dimension. On each plot, time is shown on the horizontal axis, the bottom row displays the energy feature, while the others display the filter bank features (in ascending frequency order from the bottom). All data and samples were drawn randomly.

The RNADE model has 1024 rectified-linear hidden units and a mixture of 20 one-dimensional Gaussian components per output. Given the larger scale of this dataset hyperparameter choices were again made manually using validation data, and the same minibatch training procedures for RNADE and MoG were used as for natural image patches.

The results are shown in Table 3. RNADE obtained, on average, 10 nats more per test example than a mixture of Gaussians. In Figure 2 a few examples from the test set, and samples from the MoG and RNADE models are shown. In contrast with the log-likelihood measure, there are no marked differences between the samples from each model. Both set of samples look like blurred spectrograms, but RNADE seems to capture sharper formant structures (peaks of energy at the lower frequency bands characteristic of vowel sounds).

## 5  Discussion

Mixture Density Networks (MDNs) [6] are a flexible conditional model of probability densities, that can capture skewed, heavy-tailed, and multi-modal distributions. In principle, MDNs can be applied to multi-dimensional data. However, the number of parameters that the network has to output grows quadratically with the number of targets, unless the targets are assumed independent. RNADE exploits an autoregressive framework to apply practical, one-dimensional MDNs to unsupervised density estimation.

To specify an RNADE we needed to set the parametric form for the output distribution of each MDN. A sufficiently large mixture of Gaussians can closely represent any density, but it is hard to learn the conditional densities found in some problems with this representation. The marginal for the brightness of a pixel in natural image patches is heavy tailed, closer to a Laplace distribution

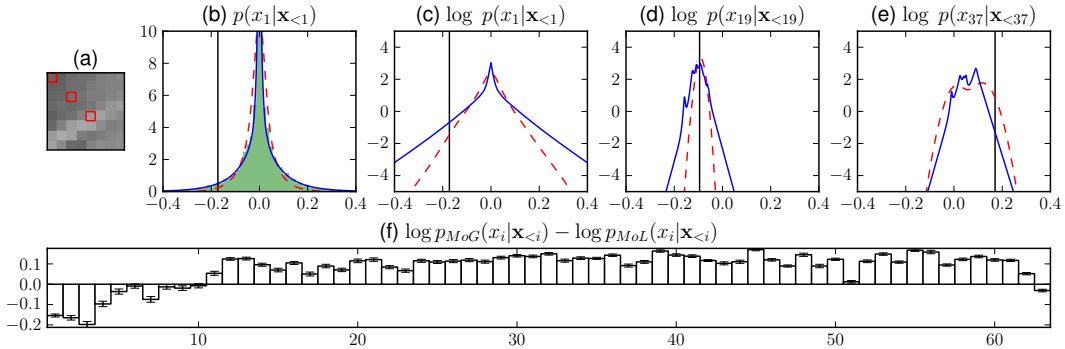

Figure 3: Comparison of Mixture of Gaussian (MoG) and Mixture of Laplace (MoL) conditionals. **(a)** Example test patch. **(b)** Density of $p(x_1)$ under RNADE-MoG (dashed-red) and RNADE-MoL (solid-blue), both with $K = 10$. RNADE-MoL closely matches a histogram of brightness values from patches in the test-set (green). The vertical line indicates the value in (a). **(c)** Log-density of the distributions in (b). **(d)** Log-density of MoG and MoL conditionals of pixel 19 in (a). **(e)** Log-density of MoG and MoL conditionals of pixel 37 in (a). **(f)** Difference in predictive log-density between MoG and MoL conditionals for each pixel, averaged over 10,000 test patches.

than Gaussian. Therefore, RNADE-MoG must fit predictions of the first pixel, $p(x_1)$, with several Gaussians of different widths, that coincidentally have zero mean. This solution can be difficult to fit, and RNADE with a mixture of Laplace outputs predicted the first pixel of image patches better than with a mixture of Gaussians (Figure 3b and c). However, later pixels were predicted better with Gaussian outputs (Figure 3f); the mixture of Laplace model is not suitable for predicting with large contexts. For image patches, a scale mixture can work well [11], and could be explored within our framework. However for general applications, scale mixtures within RNADE would be too restrictive (e.g., $p(x_1)$ would be zero-mean and unimodal). More flexible one-dimensional forms may aid RNADE to generalize better for different context sizes and across a range of applications.

One of the main drawbacks of RNADE, and of neural networks in general, is the need to decide the value of several training hyperparameters. The gradient descent learning rate can be adjusted automatically using, for example, the techniques developed by Schaul et al. [30]. Also, methods for choosing hyperparameters more efficiently than grid search have been recently developed [31, 32]. These, and several other recent improvements in the neural network field, like dropouts [33], should be directly applicable to RNADE, and possibly obtain even better performance than shown in this work. RNADE makes it relatively straight-forward to translate advances in the neural-network field into better density estimators, or at least into new estimators with different inductive biases.

In summary, we have presented RNADE, a novel 'black-box' density estimator. Both likelihood computation time and the number of parameters scale linearly with the dataset dimensionality. Generalization across a range of tasks, representing arbitrary feature vectors, image patches, and auditory spectrograms is excellent. Performance on image patches was close to a recently reported state-of-the-art mixture model [3], and RNADE outperformed mixture models on all other datasets considered.

### Acknowledgments

We thank John Bridle, Steve Renals, Amos Storkey, and Daniel Zoran for useful interactions.

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
