[Supplementary Material · supplement.pdf]

# RNADE: The real-valued neural autoregressive density-estimator
# Supplementary material

**Benigno Uria** and **Iain Murray**
School of Informatics
University of Edinburgh
{b.uria,i.murray}@ed.ac.uk

**Hugo Larochelle**
Département d'informatique
Université de Sherbrooke
hugo.larochelle@usherbrooke.ca

In this document we provide pseudo-code for the calculation of densities and learning gradients. No new material is presented. A Python implementation of RNADE is available from http://www.benignouria.com/en/research/RNADE.

## 1 Density estimation

In Algorithm 1 we detail the pseudocode for calculating the density of a datapoint under an RNADE with mixture of Gaussian conditionals. The model has parameters: $\boldsymbol{\rho} \in \mathbb{R}^D$, $\boldsymbol{W} \in \mathbb{R}^{H \times D-1}$, $\boldsymbol{c} \in \mathbb{R}^H$, $\boldsymbol{b}^\alpha \in \mathbb{R}^{D \times K}$, $\boldsymbol{V}^\alpha \in \mathbb{R}^{D \times H \times K}$, $\boldsymbol{b}^\mu \in \mathbb{R}^{D \times K}$, $\boldsymbol{V}^\mu \in \mathbb{R}^{D \times H \times K}$, $\boldsymbol{b}^\sigma \in \mathbb{R}^{D \times K}$, $\boldsymbol{V}^\sigma \in \mathbb{R}^{D \times H \times K}$

---
**Algorithm 1** Computation of $p(\boldsymbol{x})$

$\boldsymbol{a} \leftarrow \boldsymbol{c}$
$p(\boldsymbol{x}) \leftarrow 1$
**for** $d$ from 1 to $D$ **do**
    $\boldsymbol{\psi}_d \leftarrow \rho_d \boldsymbol{a}$                                       ▷ Rescaling factors
    $\boldsymbol{h}_d \leftarrow \boldsymbol{\psi}_d \, \mathbf{1}_{\boldsymbol{\psi}_d > 0}$                          ▷ Rectified linear units
    $\boldsymbol{z}_d^\alpha \leftarrow {\boldsymbol{V}_d^\alpha}^\top \boldsymbol{h}_d + \boldsymbol{b}_d^\alpha$
    $\boldsymbol{z}_d^\mu \leftarrow {\boldsymbol{V}_d^\mu}^\top \boldsymbol{h}_d + \boldsymbol{b}_d^\mu$
    $\boldsymbol{z}_d^\sigma \leftarrow {\boldsymbol{V}_d^\sigma}^\top \boldsymbol{h}_d + \boldsymbol{b}_d^\sigma$
    $\boldsymbol{\alpha}_d \leftarrow \mathrm{softmax}(\boldsymbol{z}_d^\alpha)$                        ▷ Enforce constraints
    $\boldsymbol{\mu}_d \leftarrow \boldsymbol{z}_d^\mu$
    $\boldsymbol{\sigma}_d \leftarrow \exp(\boldsymbol{z}_d^\sigma)$
    $p(\boldsymbol{x}) \leftarrow p(\boldsymbol{x}) p_{MoG}(x_d; \boldsymbol{\alpha}_d, \boldsymbol{\mu}_d, \boldsymbol{\sigma}_d)$      ▷ $p_{MoG}$ is the density of a mixture of Gaussians
    $\boldsymbol{a} \leftarrow \boldsymbol{a} + x_d \boldsymbol{W}_{\cdot,d}$         ▷ Activations are calculated recursively, $x_d$ is a scalar
**end for**
    **return** $p(\boldsymbol{x})$

---

## 2 Learning gradients

Training of an RNADE model can be done using a gradient ascent algorithm on the loglikelihood of the model given the training data. Gradients can be calculated using automatic differentiation libraries (e.g. Theano [1]). However we found our manual implementation to work faster in practice, possibly due to our recomputation of the $\boldsymbol{a}$ terms in the second *for* loop in Algorithm 2, which is more cache-friendly than storing them during the first loop.

Here we show the derivation of the gradients of each paramater of a NADE model with MoG conditionals. Following [2], we define $\phi_i(x_d \,|\, \boldsymbol{x}_{<d})$ as the density of $x_d$ under the $i$-th component of

the conditional:

$$\phi_i(x_d \,|\, \boldsymbol{x}_{<d}) = \frac{1}{\sqrt{2\pi}\boldsymbol{\sigma}_{d,i}} \exp\left\{ -\frac{(x_d - \boldsymbol{\mu}_{d,i})^2}{2\boldsymbol{\sigma}_{d,i}^2} \right\}, \tag{1}$$

and $\pi_i(x_d \,|\, \boldsymbol{x}_{<d})$ as the "responsability" of the $i$-th component for $x_d$:

$$\pi_i(x_d \,|\, \boldsymbol{x}_{<d}) = \frac{\boldsymbol{\alpha}_{d,i}\phi_i(x_d \,|\, \boldsymbol{x}_{<d})}{\sum_{j=1}^K \boldsymbol{\alpha}_{d,j}\phi_j(x_d \,|\, \boldsymbol{x}_{<d})}. \tag{2}$$

It is easy to find just by taking their derivatives that:

$$\frac{\partial p(\boldsymbol{x})}{\partial \boldsymbol{z}_{d,i}^\alpha} = \pi_i(x_d \,|\, \boldsymbol{x}_{<d}) - \boldsymbol{\alpha}_{d,i} \tag{3}$$

$$\frac{\partial p(\boldsymbol{x})}{\partial \boldsymbol{z}_{d,i}^\mu} = \pi_i(x_d \,|\, \boldsymbol{x}_{<d})\frac{x_d - \boldsymbol{\mu}_{d,i}}{\boldsymbol{\sigma}_{d,i}^2} \tag{4}$$

$$\frac{\partial p(\boldsymbol{x})}{\partial \boldsymbol{z}_{d,i}^\sigma} = \pi_i(x_d \,|\, \boldsymbol{x}_{<d})\left\{ \frac{(x_d - \boldsymbol{\mu}_{d,i})^2}{\boldsymbol{\sigma}_{d,i}^2} - 1 \right\} \tag{5}$$

Using the chain rule we can calculate the derivative of the parameters of the output layer parameters:

$$\frac{\partial p(\boldsymbol{x})}{\partial \boldsymbol{V}_d^\alpha} = \frac{\partial p(\boldsymbol{x})}{\partial \boldsymbol{z}_{d,i}^\alpha}\frac{\partial \boldsymbol{z}_{d,i}^\alpha}{\boldsymbol{V}_d^\alpha} = \frac{\partial p(\boldsymbol{x})}{\partial \boldsymbol{z}_{d,i}^\alpha}\boldsymbol{h} \tag{6}$$

$$\frac{\partial p(\boldsymbol{x})}{\partial \boldsymbol{b}_d^\alpha} = \frac{\partial p(\boldsymbol{x})}{\partial \boldsymbol{z}_{d,i}^\alpha}\frac{\partial \boldsymbol{z}_{d,i}^\alpha}{\boldsymbol{b}_d^\alpha} = \frac{\partial p(\boldsymbol{x})}{\partial \boldsymbol{z}_{d,i}^\alpha} \tag{7}$$

$$\frac{\partial p(\boldsymbol{x})}{\partial \boldsymbol{V}_d^\mu} = \frac{\partial p(\boldsymbol{x})}{\partial \boldsymbol{z}_{d,i}^\mu}\frac{\partial \boldsymbol{z}_{d,i}^\alpha}{\boldsymbol{V}_d^\mu} = \frac{\partial p(\boldsymbol{x})}{\partial \boldsymbol{z}_{d,i}^\mu}\boldsymbol{h} \tag{8}$$

$$\frac{\partial p(\boldsymbol{x})}{\partial \boldsymbol{b}_d^\mu} = \frac{\partial p(\boldsymbol{x})}{\partial \boldsymbol{z}_{d,i}^\mu}\frac{\partial \boldsymbol{z}_{d,i}^\alpha}{\boldsymbol{b}_d^\mu} = \frac{\partial p(\boldsymbol{x})}{\partial \boldsymbol{z}_{d,i}^\mu} \tag{9}$$

$$\frac{\partial p(\boldsymbol{x})}{\partial \boldsymbol{V}_d^\sigma} = \frac{\partial p(\boldsymbol{x})}{\partial \boldsymbol{z}_{d,i}^\sigma}\frac{\partial \boldsymbol{z}_{d,i}^\alpha}{\boldsymbol{V}_d^\sigma} = \frac{\partial p(\boldsymbol{x})}{\partial \boldsymbol{z}_{d,i}^\sigma}\boldsymbol{h} \tag{10}$$

$$\frac{\partial p(\boldsymbol{x})}{\partial \boldsymbol{b}_d^\sigma} = \frac{\partial p(\boldsymbol{x})}{\partial \boldsymbol{z}_{d,i}^\sigma}\frac{\partial \boldsymbol{z}_{d,i}^\alpha}{\boldsymbol{b}_d^\sigma} = \frac{\partial p(\boldsymbol{x})}{\partial \boldsymbol{z}_{d,i}^\sigma} \tag{11}$$

By "backpropagating" the we can calculate the partial derivatives with respect to the output of the hidden units:

$$\frac{\partial p(\boldsymbol{x})}{\partial \boldsymbol{h}_d} = \frac{\partial p(\boldsymbol{x})}{\partial \boldsymbol{z}_{d,i}^\alpha}\frac{\partial \boldsymbol{z}_{d,i}^\alpha}{\partial \boldsymbol{h}_d} + \frac{\partial p(\boldsymbol{x})}{\partial \boldsymbol{z}_{d,i}^\mu}\frac{\partial \boldsymbol{z}_{d,i}^\mu}{\partial \boldsymbol{h}_d} + \frac{\partial p(\boldsymbol{x})}{\partial \boldsymbol{z}_{d,i}^\sigma}\frac{\partial \boldsymbol{z}_{d,i}^\sigma}{\partial \boldsymbol{h}_d} \tag{12}$$

$$= \frac{\partial p(\boldsymbol{x})}{\partial \boldsymbol{z}_{d,i}^\alpha}\boldsymbol{V}_d^\alpha + \frac{\partial p(\boldsymbol{x})}{\partial \boldsymbol{z}_{d,i}^\mu}\boldsymbol{V}_d^\mu + \frac{\partial p(\boldsymbol{x})}{\partial \boldsymbol{z}_{d,i}^\sigma}\boldsymbol{V}_d^\sigma \tag{13}$$

and calculate the partial derivatives with respect to all other parameters in RNADE:

$$\frac{\partial p(\boldsymbol{x})}{\partial \boldsymbol{\psi}_d} = \frac{\partial p(\boldsymbol{x})}{\partial \boldsymbol{h}_d}\mathbf{1}_{\psi_d>0} \tag{14}$$

$$\frac{\partial p(\boldsymbol{x})}{\partial \rho_d} = \sum_j \frac{\partial p(\boldsymbol{x})}{\partial \boldsymbol{\psi}_{d,j}}\boldsymbol{a}_{d,j} \tag{15}$$

$$\frac{\partial p(\boldsymbol{x})}{\partial \boldsymbol{a}_d} = \frac{\partial p(\boldsymbol{x})}{\partial \boldsymbol{a}_{d+1}} + \frac{\partial p(\boldsymbol{x})}{\partial \boldsymbol{h}_d}\rho_d\mathbf{1}_{\psi_d>0} \tag{16}$$

$$\frac{\partial p(\boldsymbol{x})}{\partial \boldsymbol{W}_{\cdot,d}} = \frac{\partial p(\boldsymbol{x})}{\partial \boldsymbol{a}_d}x_d \tag{17}$$

$$\frac{\partial p(\boldsymbol{x})}{\partial \boldsymbol{c}} = \frac{\partial p(\boldsymbol{x})}{\partial \boldsymbol{a}_1} \tag{18}$$

Note that gradients are calculated recursively, due to (16), starting at $d = D$ and progressing down to $d = 1$.

---

**Algorithm 2** Computation of the learning gradients for a datapoint $\boldsymbol{x}$

---

$\boldsymbol{a} \leftarrow \boldsymbol{c}$
**for** $d$ from 1 to $D$ **do** &emsp;&emsp;&emsp;&emsp;&emsp;&emsp;&emsp;&emsp;&emsp; ▷ Compute the activation of the last dimension
&emsp; $\boldsymbol{a} \leftarrow \boldsymbol{a} + x_d \boldsymbol{W}_{\cdot,d}$
**end for**
**for** $d$ from $D$ to 1 **do** &emsp;&emsp;&emsp;&emsp;&emsp;&emsp;&emsp;&emsp;&emsp;&emsp;&emsp; ▷ Backpropagate errors
&emsp; $\boldsymbol{\psi} \leftarrow \rho_d \boldsymbol{a}$ &emsp;&emsp;&emsp;&emsp;&emsp;&emsp;&emsp;&emsp;&emsp;&emsp;&emsp;&emsp;&emsp;&emsp; ▷ Rescaling factors
&emsp; $\boldsymbol{h} \leftarrow \boldsymbol{\psi} \, \mathbf{1}_{\boldsymbol{\psi} > 0}$ &emsp;&emsp;&emsp;&emsp;&emsp;&emsp;&emsp;&emsp;&emsp;&emsp;&emsp; ▷ Rectified linear units
&emsp; $\boldsymbol{z}^\alpha \leftarrow \boldsymbol{V}_d^{\alpha\top} \boldsymbol{h} + \boldsymbol{b}_d^\alpha$
&emsp; $\boldsymbol{z}^\mu \leftarrow \boldsymbol{V}_d^{\mu\top} \boldsymbol{h} + \boldsymbol{b}_d^\mu$
&emsp; $\boldsymbol{z}^\sigma \leftarrow \boldsymbol{V}_d^{\sigma\top} \boldsymbol{h}_d + \boldsymbol{b}_d^\sigma$
&emsp; $\boldsymbol{\alpha} \leftarrow \mathrm{softmax}(\boldsymbol{z}^\alpha)$ &emsp;&emsp;&emsp;&emsp;&emsp;&emsp;&emsp;&emsp; ▷ Enforce constraints
&emsp; $\boldsymbol{\mu} \leftarrow \boldsymbol{z}^\mu$
&emsp; $\boldsymbol{\sigma} \leftarrow \exp(\boldsymbol{z}^\sigma)$
&emsp; $\boldsymbol{\phi} \leftarrow \frac{1}{2} \frac{(\boldsymbol{\mu} - \boldsymbol{x}_d)^2}{\boldsymbol{\sigma}^2} - \log \boldsymbol{\sigma} - \frac{1}{2}\log(2\pi)$ &emsp;&emsp; ▷ Calculate gradients
&emsp; $\boldsymbol{\pi} \leftarrow \frac{\boldsymbol{\alpha}\boldsymbol{\phi}}{\sum_{j=1}^{K} \boldsymbol{\alpha}_j \boldsymbol{\phi}_j}$
&emsp; $\partial z^\alpha \leftarrow \boldsymbol{\pi} - \boldsymbol{\alpha}$
&emsp; $\partial \boldsymbol{V}_d^\alpha \leftarrow \partial z^\alpha \boldsymbol{h}$
&emsp; $\partial \boldsymbol{b}_d^\alpha \leftarrow \partial z^\alpha$
&emsp; $\partial z^\mu \leftarrow \boldsymbol{\pi}(x_d - \boldsymbol{\mu})/\boldsymbol{\sigma}^2$
&emsp; $\partial z^\mu \leftarrow \partial z^\mu * \sigma$ &emsp;&emsp;&emsp; ▷ Move tighter components slower, allows higher learning rates
&emsp; $\partial \boldsymbol{V}_d^\mu \leftarrow \partial z^\mu \boldsymbol{h}$
&emsp; $\partial \boldsymbol{b}_d^\mu \leftarrow \partial z^\mu$
&emsp; $\partial z^\sigma \leftarrow \boldsymbol{\pi}\{(x_d - \boldsymbol{\mu})^2/\boldsymbol{\sigma}^2 - 1\}$
&emsp; $\partial \boldsymbol{V}_d^\sigma \leftarrow \partial z^\sigma \boldsymbol{h}$
&emsp; $\partial \boldsymbol{b}_d^\sigma \leftarrow \partial z^\sigma$
&emsp; $\partial \boldsymbol{h} \leftarrow \partial \boldsymbol{z}^\alpha \boldsymbol{V}_d^\alpha + \partial \boldsymbol{z}^\mu \boldsymbol{V}_d^\mu + \partial \boldsymbol{z}^\sigma \boldsymbol{V}_d^\sigma$
&emsp; $\partial \boldsymbol{\psi} \leftarrow \partial \boldsymbol{h} \mathbf{1}_{\boldsymbol{\psi} > 0}$ &emsp;&emsp;&emsp; ▷ Second factor: indicator function with condition $\boldsymbol{\psi} > 0$
&emsp; $\partial \rho_d \leftarrow \sum_j \partial \boldsymbol{\psi}_j a_j$
&emsp; $\partial \boldsymbol{a} \leftarrow \partial \boldsymbol{a} + \partial \boldsymbol{\psi} \rho$
&emsp; $\partial \boldsymbol{W}_{\cdot,d} \leftarrow \partial \boldsymbol{a} x_d$
&emsp; **if** $d = 1$ **then**
&emsp;&emsp; $\partial \boldsymbol{c} \leftarrow \partial \boldsymbol{a}$
&emsp; **else**
&emsp;&emsp; $\boldsymbol{a} \leftarrow \boldsymbol{a} - x_d \boldsymbol{W}_{\cdot,d}$
&emsp; **end if**
**end for**
&emsp; **return** $\partial \boldsymbol{\rho}, \partial \boldsymbol{W}, \partial \boldsymbol{c}, \partial \boldsymbol{b}^\alpha, \partial \boldsymbol{V}^\alpha, \partial \boldsymbol{b}^\mu, \partial \boldsymbol{V}^\mu, \partial \boldsymbol{b}^\sigma, \partial \boldsymbol{V}^\sigma$

---