[Reviews · NeurIPS 2013]

Submitted by Assigned_Reviewer_4

Summary
=======
The authors present a directed model for estimating the density of continuous random variables by exploiting the chain rule of probability theory. They propose to use particular weight-sharing constraints which have proven useful for modeling discrete data and combine it with mixture density networks. They show that their model can generally outperform large mixtures of Gaussians when applied to image patches, speech signals, and several smaller datasets.


Comments
========

Weight sharing
--------------
Since the main difference to related work appears to be in the RBM-inspired weight-sharing, it would be interesting to see a more thorough investigation of its effects. While it is clear that it can reduce computational costs, its effects on performance have not been fully explored. One would expect the weight-sharing to reduce overfitting where data is scarce, and to hurt performance where data is plenty. It would therefore be interesting to see some results on the extent to which weight sharing can help and to learn more about regimes where weight sharing might actually hinder performance.


Natural image patches
---------------------
In Section 4.2, RNADE is shown to perform similarly to a large mixtures of Gaussians when applied to natural image patches. While this already represents a competitive result, I believe that RNADE could fare even better here. Because of a mixture model's inability to represent independencies efficiently, it scales poorly to high dimensional data. I therefore suspect that RNADE would outperform a mixture model when tested on slightly larger patches.

The number of training and validation points used for training RNADE also appears to be quite small. 25,000 training points (1,000 batches with 25 data points each) in my experience is not a lot, even for smaller image patches. If the performance does in fact not improve with more data (or, in other words, there is no overfitting with 25,000 training points), then this would also speak for the advantages of using the proposed weight-sharing constraints – which could be a point worth mentioning.


Related work
------------
Since several directed models for real valued data already exist which all use Gaussian mixtures to represent conditional distributions and – which may not be obvious – have similar gating mechanisms to predict the mixing weights (Domke et al., 2008; Hosseini et al., 2009; Theis et al., 2012), a comparison with at least one of these models would have been nice.


Minor comments
--------------
It would be easier to judge the size of the differences in performance in Table 2 and 3 if additional models were included in the comparison, as in Table 1.

There appears to be a $\rho_d$ missing in Equation 4.


Quality
=======
The methods used in the paper are technically sound. The authors provide extensive comparisons of their model and mixture models on several datasets and go to great lengths to ensure that the results are representative by performing 10-fold cross-validation or using very large test sets where possible.


Clarity
=======
The paper is well written and easy to follow. It also appears to include enough detail to reproduce the results.


Originality
===========
The paper explores the nontrivial extension of NADE – a model for discrete data – to the continuous case. While related models exist which also model continuous data and which share some similarities with the proposed model, there are also plenty of differences which make this an interesting and original contribution.


Significance
============
Density estimation is an important problem underlying many applications. Just like the superior density estimation performance of NADE has proven useful in solving applications involving discrete data (Larochelle & Lauly, 2012), RNADE has the potential to be useful in tackling applications involving continuous signals.
Summary: The paper presents a nontrivial adaptation of a successful directed model for discrete data to the continuous case along with extensive empirical results demonstrating a very good performance on several datasets.

Submitted by Assigned_Reviewer_5

The authors extend the Neural Autoregressive Distribution Estimator (NADE) to
perform density estimation for real-valued vectors. The main difference from
NADE is modeling the conditional distribution of the next vector element given
the preceding ones with a mixture density network instead of logistic
regression. Typically the distributions being mixed are univariate Gaussians,
the mean and stdev. of which are functions of the hidden layer activations. As
in NADE, the inputs to the hidden units are computed efficiently in time linear
in the input dimensionality.

This is a nicely written paper, based on a simple idea that seems to work well.
The experimental section is thorough and convincing at showing that RNADE is
a good general-purpose density model.

Figure 2 caption claims that the samples came from an RNADE model with 512
hidden units, while in Section 4.3 the model is said to have 1024 units. Is this
a typo or are these actually different models?

Finally, though mixtures of Gaussians are a reasonable baseline, it would be
interesting to see how RNADE compares to something more distributed, such
as FVSBN-like density models.
Summary: A nicely written paper based on a simple idea that seems to work well.

Submitted by Assigned_Reviewer_8

This paper proposes a simple yet effective model called RNADE for joint density estimation for real-valued vectors. The method is well motivated and clearly presented. Extensive experiments show that the RNADE outperforms many other approaches on various datasets.

While the quality of this paper is good, I have some concerns regarding the significance of the paper. First, the proposed method is a generalization of the NADE method, so the novelty is not significant. Second, in most of modern learning and inference tasks, it is not necessary to have an accurate estimation of density values. It would be great if some results on high-level tasks can be shown; for example classification or de-noising.
Summary: The idea in this paper is well motivated, clearly presented, and well justified. But the significance and novelty of the method is questionable.
Author Feedback

Author rebuttal: We would like to thank the reviewers for their helpful criticism. In short, the two typos would be corrected, and a comparison with a FVBN-like model would be added to Table 1 (our model performs better) in a camera-ready version. A more detailed answer to each of the issues raised by the reviewers follows.

The 2 typos identified by reviewer 1 (missing $\rho_d$ in equation 4) and reviewer 2 (number of units per layer in the caption of Figure 2) are indeed mistakes that would be corrected in a camera-ready version.

Reviewer 1 recommends using a more extensive training dataset for image patches, considering 25000 datapoints not enough. Our training procedure uses 25000 datapoints (sampled with replacement from a pool of more than 20 million) _per epoch_ (of which we do 200) totalling 5 million (see lines 257-258 in the paper).

Reviewer 1 suggests testing the performance of RNADE on bigger image patches, saying it is likely it would beat MoGs. We agree. As can be shown in Figure 3, the first few pixels (and those near the edges of the patch) are the most difficult to predict for RNADE. On a bigger patch the proportion of those pixels is smaller and therefore we would expect it to obtain a higher loglikelihood per pixel. Whereas MoG models show a slight decrease in loglikelihood per pixel for bigger patches (as shown in Zoran ans Weiss' NIPS 2012 paper). Still, the purpose of our paper was not beating the state-of-the-art on image patches modelling, but to show that RNADE is a generally capable model. A comparison of the two model across different patch sizes should certainly be included in a more specialized image modelling paper.

Reviewer 1 also recommends a comparison with either Hosseni, Domke or Theis. The goal of our paper is showing RNADE is a flexible general purpose model for real-valued data. A thorough comparison with the more specialized full-image modelling literature would have taken much space. We considered more interesting to report the performance of RNADE in other domains like speech acoustics. However, in the second paragraph of our discussion section, we agree that a Gaussian scale mixture approach, followed in Theis et al, may be a better option than a mixture of Gaussians for natural image patches (see line 375 in the paper).

Reviewers 1 and 2 raise a good point, may RNADE's weight sharing hinder performance in any case? A comparison with a system where no weight sharing is used would certainly be interesting. However, a system of that kind would be impractical in high-dimensional datasets (like speech acoustics or image patches). We have run experiments using a FVBN-like model (with a MoG MDN top layer) on the lower-dimensional datasets of section 4.1. The results are inferior to RNADE. We would report these results in the camera ready version.

Reviewer 3 while acknowledging the good quality of our paper, judges the innovation in it (and indeed the topic of density estimation) as not very relevant. We consider the extension of NADE to real-valued data an important contribution. The authors are not aware of any general purpose, tractable models of real-valued data able to compete in performance with MFAs (on big datasets). Even some of the most popular intractable models (like Gaussian RBMs) offer very poor results. The introduction of a tractable and capable density estimator opens possibilities to practitioners like the use of Bayes' classifiers (by comparing class likelihoods), working with missing data (see, for example, the missing feature literature on noisy speech recognition, where both data imputation and marginalisation are used) or its use to generate data (as in speech synthesis, and image inpainting). No doubt results on high-level tasks are interesting, but we considered more important to report test-likelihoods and samples, given that our model is a density estimator and one of its main advantages is its tractability. We are exploring the use of RNADE in high-level tasks, and will report its results at specialized venues.